# Beyond Averages: Unpacking Disparities in School-Based Vaccination Coverage in Eastern Sydney: An Ecological Analysis

**DOI:** 10.3390/vaccines12080888

**Published:** 2024-08-05

**Authors:** Leigh McIndoe, Elizabeth Wilson, Mark J. Ferson, Vicky Sheppeard

**Affiliations:** 1Public Health Unit, South Eastern Sydney Local Health District, Sydney, NSW 2031, Australia; leigh.mcindoe@health.nsw.gov.au (L.M.); mark.ferson@health.nsw.gov.au (M.J.F.); 2New South Wales Ministry of Health, Sydney, NSW 2065, Australia; 3School of Population Health, University of New South Wales, Sydney, NSW 2050, Australia; 4School of Public Health, University of Sydney, Camperdown, NSW 2050, Australia

**Keywords:** school-based immunisation, adolescent vaccination, HPV vaccine, dTpa vaccine, immunisation coverage, differential uptake, quasi-Poisson regression

## Abstract

School vaccination programs are crucial for achieving high immunisation coverage among adolescents, but substantial disparities exist across schools and regions. This ecological study aimed to determine associations between school characteristics and vaccination coverage for diphtheria–tetanus–acellular pertussis (dTpa) and human papillomavirus (HPV) vaccines among year 7 students in southeastern Sydney. An analysis of data from 70 mainstream schools participating in the 2019 South Eastern Sydney Local Health District School Vaccination Program utilised quasi-Poisson regression models to assess associations between vaccination coverage and school attendance, socio-educational status, Aboriginal enrolments, language background other than English (LBOTE), school sector (government, Catholic, or independent), and coeducation status. Median school coverage was 88% for dTpa, 88% for HPV—girls, and 86% for HPV—boys, with interquartile ranges of 82–93%, 84–92%, and 78–91%, respectively. Higher school attendance was associated with increased dTpa vaccination coverage (PR 1.14, 95% CI 1.02–1.27). Single-sex schools showed higher HPV vaccination coverage compared to coeducational schools for both girls (PR 2.24, 95% CI 2.04–2.46) and boys (PR 1.89, 95% CI 1.72–2.08). No significant associations were found for ICSEA, Aboriginal enrolments, LBOTE, or school sector. School attendance and coeducational status significantly influenced vaccination coverage, with differential impacts on dTpa and HPV vaccines. These findings highlight the need for targeted strategies to address disparities in school-based vaccination programs. Research using qualitative methods could be useful to understand the beliefs and attitudes contributing to these disparities in vaccine uptake so that programs can be tailored to maximise participation.

## 1. Introduction

School vaccination programs have proven highly effective in achieving widespread immunisation coverage among adolescents. These programs are crucial for maximising protection against future disease risk, bolstering waning immunity, and reaching individuals who may have missed childhood vaccinations [1,2,3]. Additionally, school-based vaccination has been shown to reduce the burden of disease not only within the vaccinated population but also in the broader community [2].

The effectiveness of schools as settings for vaccination programs is widely recognised, driven by their ability to reach a substantial number of children and adolescents during mandatory attendance years and offer convenience to families, overcoming common barriers associated with primary care delivery, such as cost and accessibility [2,4]. Furthermore, school environments nurture peer support, establish social norms, and play a pivotal role in reducing disparities in vaccination access [5].

In Australia, the National Immunisation Program (NIP), funded by the Australian government and implemented by state and territory departments of health, has routinely offered a booster dose of diphtheria–tetanus–acellular pertussis (dTpa) vaccination to adolescents aged 12 to 13 years in secondary school (year 7 or 8) since 2010 [3,6,7,8]. Human papillomavirus (HPV) vaccination was introduced as a 3-dose schedule for girls in year 7 in 2007 and extended to boys in 2013 [3,6,7,8]. In 2017, the HPV vaccination transitioned to a two-dose schedule before eventually shifting to a single dose in 2023, aligning with World Health Organization recommendations [9]. Additionally, in 2017, Australia launched a meningococcal ACWY vaccination (MenACWY) program targeting year 11 and 12 students, shifting to year 10 and 11 students in 2018 before routine provision for year 10 students (14–16 years) since 2019 [6,7]. All vaccines available through the NIP are free to the recipients; however, outside of school vaccination programs, a provider fee may be charged.

Despite the recognised effectiveness of school-based immunisation programs, coverage data reveal substantial differences across Australian states and considerable regional variation [1,7,8,10,11,12,13]. Significant variation has also been reported internationally. For instance, in Canada, disparities in HPV uptake were reported between provinces [14]. Similarly, Scotland has witnessed varying uptake rates for the school-based vaccine booster for diphtheria, tetanus, and polio (Td/IPV) and meningococcal (MenACWY) doses across regions [15].

In the state of New South Wales (NSW), public health units are responsible for implementing the school vaccination program, including data collection and reporting [6,10]. Within this framework, a dedicated team of nurses and support staff from the South Eastern Sydney Local Health District (SESLHD) Public Health Unit implements the program in secondary schools located in the area. They collaborate closely with schools to coordinate vaccination clinics and secure parental or guardian consent.

The SESLHD School Vaccination Program has a strong track record of achieving high vaccine coverage, surpassing the state average for dTpa and HPV vaccines since 2012 [10]. However, in our experience while overall coverage rates in SESLHD are high, these figures obscure lower vaccination rates in certain schools where coverage is considerably below the state average.

While few studies have delved into the disparities in uptake of school-based vaccination, previous research at the individual level indicates associations between adolescent immunisation coverage and factors such as socio-economic status, ethnicity, and demographic characteristics [15,16]. SESLHD school program data reveal variations in coverage levels even among schools within the same suburbs, suggesting multiple contributing factors. This is corroborated by data from England, where COVID-19 vaccination rates among students attending state-funded secondary schools exhibited greater variation between schools within the same region than between regions [17].

To better understand the factors contributing to these observed disparities in vaccination coverage, there is a need to investigate more nuanced measures specific to the school environment, beyond the aggregated socio-economic conditions and deprivation indices, such as the school-postcode-based Socio-Economic Indexes for Areas (SEIFA) of disadvantage used in previous research. In the Australian context, the Index of Community Socio-Educational Advantage (ICSEA) provides a composite measure assessing the relative socio-educational advantage or disadvantage of schools [18]. A score is generated for each school in Australia that enrols children in years 1–10. It is derived from the level of educational attainment and occupation of parents of students enrolled at the school, the remoteness of the school from capital cities, and the proportion of students identifying as indigenous. The median ICSEA value is 1000, with a standard deviation of 100. Extremely socio-educationally disadvantaged schools would typically have an ICSEA score around 500, and the most socio-educationally advantaged schools have a score around 1300 [18]. The ICSEA score has not previously been explored as a factor in vaccination coverage.

To address these research gaps, we aimed to determine if any measurable school characteristics, including ICSEA, are associated with variation in dTpa and HPV vaccine uptake between schools and identify where we can best target strategies to improve uptake in low-coverage schools.

## 2. Methods

### 2.1. Setting

The NSW Health service is structured into 15 local health districts, each responsible for delivering healthcare services within defined geographical regions [4]. SESLHD is one of the largest metropolitan districts, covering a significant area of 468 square kilometres. SESLHD provides healthcare services to a population of over 991,000 residents, representing 13.3% of the state. The population is diverse, with 53% of residents born overseas, 34% of households speaking a non-English language, and 1% identifying as Aboriginal and/or Torres Strait Islander [19].

### 2.2. Study Design and Study Population

We conducted an ecological study, with schools as the primary unit of analysis. The study population comprised mainstream secondary schools located within SESLHD boundaries, with more than 10 students enrolled in year 7 (typically 11–12 years of age). A small subset of schools following different immunisation schedules were excluded from the analysis, including two intensive English schools, one international school, and one special needs school.

### 2.3. Data Collation and Measures

Vaccination data for the HPV and dTpa vaccines for year 7 students were recorded by school immunisation teams and collated by public health units and NSW Health. Vaccination and enrolment data for the 2019 calendar year were obtained from the NSW School Immunisation Program database, maintained by the SESLHD Public Health Unit. These records were integrated with a dataset provided by the Australian Curriculum, Assessment, and Reporting Authority (ACARA), using the ACARA ID as the linking variable.

We examined three outcomes of interest: dTpa vaccination coverage, HPV vaccination coverage in girls, and HPV vaccination coverage in boys. For HPV vaccination coverage, initiation coverage (dose 1) was chosen over completion to reflect the schedule change in 2023. Coverage for each outcome was calculated using the counts of students vaccinated as the numerator and the school population enrolments for year 7 at the start of the year as the denominator. The year 7 vaccination coverage estimates do not include any school catch-up vaccinations given in year 8 in the following year. A detailed description of the coverage calculations is available elsewhere [11].

We examined the association between these outcomes and various factors that were chosen a priori based on the literature and consultation with SESLHD school immunisation coordinators [1,13,14,15,16]. These factors included the following:School attendance, defined as the percentage of days attended in semester 1 and classified as equal to or less than 92% or greater than 92%;Coeducation status (single-sex or coeducational);School sector (government, Catholic, or independent);The percentage of students enrolled at the school who speak a language other than English (LBOTE) at home (classified as equal to or less than 25% or greater than 25%);The percentage of students enrolled at the school who are of Aboriginal and/or of Torres Strait Islander descent (classified as equal to or less than 1% or greater than 1%);The ICSEA score for the school (classified as high (≥1100), medium (1000–1099), and low (≤999)).

The chosen cut-off points for these classifications were based on relevant benchmarks and statistics. For instance, the attendance classification was based on the average year 7 attendance across NSW in 2019 [20]. The classification of LBOTE was based on the percentage of the NSW population who spoke a language other than English at home in 2019 [21], and the classification for Aboriginal and/or Torres Strait Islander descent was based on the percentage of individuals living in SESLHD [19].

### 2.4. Statistical Analysis

We calculated HPV vaccination coverage for girls and boys separately as well as overall dTpa vaccination coverage for each school and described vaccination coverage according to each characteristic of interest.

We used quasi-Poisson regression models to assess associations between vaccination coverage for each outcome (dTpa, HPV in girls, and HPV in boys) and school characteristics. To model school-level vaccination coverage, the observed number of vaccinated students was included as the outcome variable, and the natural logarithm of the observed number of students in year 7 was included as an offset term. School characteristics were included as explanatory variables. Models were tested for goodness of fit using analysis of deviance and pseudo R^2^ statistics. Adjusted prevalence ratios (PRs) were calculated from the quasi-Poisson models with corresponding 95% confidence intervals (95% CI).

Analyses were performed using R Statistical Software (version 4.1.1 for Windows; R Foundation for Statistical Computing, Vienna, Austria, http://www.r-project.org/), including RStudio (desktop version 2023.12.1.402; RStudio: Integrated Development for R, Boston, USA, http://www.rstudio.com/). The ‘glm’ function in the stats package was used to develop models [22]. Quasi-Poisson regression models were used due to the existence of overdispersion. This was measured using the ‘dispersiontest’ function in the AER package [23].

## 3. Results

Complete data were available for all 72 mainstream schools participating in the SESLHD school vaccination program in 2019. Two schools with enrolments of less than 10 students were excluded. Collectively, the 70 schools included in the analysis had a total enrolment of 9379 students in year 7 (range 14–251 per school). This included data for 70 schools for dTpa, 52 schools that enrolled boys for HPV, and 56 schools that enrolled girls for HPV. Most were government schools (47.1%), and most were coeducational (52.9%). Median attendance was high (93.0%), and only 10% of schools had a low ICSEA score. Students came from diverse backgrounds, with a median of 30.5% speaking a language other than English at home. The median percentage of students who were of Aboriginal and/or Torres Strait Islander descent was 1% (Table 1).

The median school-level vaccination coverage was 88.2% for dTpa, 88.4% for HPV in girls, and 86.3% for HPV in boys. We found substantial variability, with dTpa vaccination coverage ranging from 31.3% to 98.0%, while HPV vaccination coverage ranged from 25.0% to 98.7% for girls and from 35.3% to 97.4% for boys. For each vaccine, coverage was highest in Catholic schools, single-sex schools, and schools with higher attendance (Table 1). The lowest coverage for each vaccine was found in a medium-sized government school.

In fully adjusted multivariable models (Table 2), there was a significant association between attendance and dTpa vaccination coverage: Students who attended schools with higher year 7 attendance rates were 14% more likely to be vaccinated for dTpa compared to students who attended schools with lower year 7 attendance rates (PR 1.14 (95% CI 1.02–1.27)). There was also a significant association between HPV vaccination coverage and coeducation status for both boys and girls: Students who attended single-sex schools were more likely to be vaccinated for HPV compared to students who attended coeducation schools (PR 2.24 (95% CI 2.04–2.46) and PR 1.89 (95% CI 1.72–2.08) for girls and boys, respectively). We did not find significant associations between vaccine coverage and ICSEA, LBOTE, Aboriginal enrolments, or school sector.

## 4. Discussion

Median vaccination coverage for each outcome (dTpa, HPV in girls, and HPV in boys) within the year 7 SESLHD cohort surpassed coverage estimates for the same vaccines in adolescent students across NSW by 2–3% [10,11]. However, even though overall school-based vaccination coverage in SESLHD is high, some disparities exist. Our study found significant associations between vaccination coverage and two key factors: school attendance and coeducation status. No significant associations were found between vaccine uptake and ICSEA, Aboriginal enrolments, LBOTE, or school sector.

While not reaching statistical significance, independent schools tended to have lower coverage of dTpa and HPV in girls. The independent sector is diverse, including schools affiliated with a range of religious faiths as well as non-religious-based organisations. While there is insufficient power to explore variation in this sector with this database, future studies could include more year cohorts or a larger geographic area to determine if some independent school types require additional support to improve coverage.

While also not statistically significant, HPV coverage in girls appears to be lower than both HPV in boys and dTpa in schools with a higher proportion of students from a language background other than English. This is in contrast to the finding of Vujovich-Dunn, who found that schools with a lower proportion of students with a LBOTE had lower uptake of HPV vaccine compared to dTpa [8]. The influence of language background is an important factor to explore in future studies to quantify its potential impact and explore initiatives to overcome any barriers to HPV uptake in girls from linguistically diverse backgrounds.

### 4.1. Attendance Rate

A key finding was the significant association between school attendance rates and dTpa vaccination coverage. While non-significant, the point estimate of association with HPV vaccination coverage in girls and boys was similar (1.14 and 1.09, respectively). The lack of a significant association between attendance and coverage for the HPV cohorts may have arisen from the smaller cohort sizes (52–56 schools, compared to 70 for dTpa cohort), resulting in wider confidence intervals.

The association between attendance and coverage aligns with the intuitive expectation that better attendance provides more consistent opportunities for students to receive school-based vaccinations. It may also indicate more consistent exposure to school-based health promotion efforts. Previous research has found significant associations between absenteeism and HPV vaccination coverage [8,24,25,26].

### 4.2. Coeducation Status

Another notable finding in our study was the substantially higher HPV vaccination coverage in single-sex schools compared to coeducational schools for both boys and girls. This aligns with previous research in Australia, such as that of Sisnowski et al., who reported significantly lower HPV vaccination completion rates in coeducational schools, particularly for girls [12].

Several factors may contribute to this disparity, including differences in health education approaches, school cultures, social dynamics, and parental attitudes. Single-sex schools might offer more tailored health education programs that emphasise the importance of HPV vaccination in the context of gender-specific health concerns. The social environment in single-sex schools may also be more conducive to HPV vaccine acceptance and uptake. Additionally, parents who choose single-sex education might have different attitudes and priorities towards health interventions.

Interestingly, we found no significant association between dTpa vaccination coverage and coeducation status. The variation between vaccines was also observed in a UK study that reported higher coverage for the MenACWY vaccine in single-sex schools but no difference in HPV coverage between mixed and female-only schools [27]. These varying results across different vaccines suggest that factors influencing vaccine uptake might be vaccine-specific, with single-sex schools potentially offering better support or emphasising certain vaccinations differently.

### 4.3. Socio-Educational Background

This study represents, to our knowledge, the first investigation of the potential impact of the Index of Community Socio-Educational Advantage (ICSEA) on vaccination coverage in schools. Our findings did not show an association between ICSEA and vaccination coverage for diphtheria–tetanus–acellular pertussis (dTpa) or human papillomavirus (HPV), suggesting relatively equal coverage across schools with different levels of socio-educational advantage. These results are consistent with some previous research, such as an ecological study that observed equitable HPV vaccination coverage across socio-economic groups and geographical areas in Australia using the SEIFA IRSD index [28]. This lack of association may reflect the success of SESLHD’s school-based program in reducing equity gaps, or it could indicate that school-level measures do not fully capture individual or household-level socio-economic disparities. Furthermore, the absence of significant differences between school sector suggests that a school’s socio-economic background may be less influential than other factors in determining vaccination coverage. This finding supports the potential for consistent implementation of vaccination programs across different school sectors, emphasising the role of well-structured and universally applied public health initiatives.

The lack of significant differences may also be due to the majority of schools in SESLHD being categorised as high- or medium-ICSEA, resulting in less variation across the socio-educational spectrum. This limited range might obscure more pronounced disparities observed in regions with greater socio-economic diversity.

While our study did not identify a significant association between ICSEA and vaccine coverage, the broader literature presents a mixed picture, indicating that the relationship between socio-economic background and vaccination coverage is multifaceted and context-dependent. For instance, contrasting results emerged from several other Australian studies examining the association between vaccine uptake and SEIFA IRSD. Vujovich-Dunn et al. reported that schools located in lower socio-economic areas were associated with lower HPV vaccination coverage [8]. Similarly, another study conducted in Western Australia found a consistently negative association between HPV vaccination coverage and socio-economically disadvantaged areas [29]. However, this same study did not find an association between dTpa vaccination coverage and socio-economic disadvantage, highlighting the complexity of factors influencing vaccine uptake.

### 4.4. Strengths and Limitations

This study offers several strengths, including comprehensive coverage of mainstream schools in SESLHD and the use of robust statistical methods to assess associations between school characteristics and vaccination coverage. The novel application of ICSEA provides a unique perspective on socio-educational impacts, while the examination of both dTpa and HPV vaccination coverage allows for comparisons between different vaccines, revealing vaccine-specific patterns in coverage disparities.

However, several limitations should be acknowledged. Despite identifying significant associations, analyses of deviance and pseudo R^2^ measures suggest that other unaccounted factors may explain additional variation in vaccination coverage. HPV vaccine uptake in particular is likely influenced by broader factors such as beliefs and values, particularly in relation to adolescent sexual activity, concerns about safety, and misinformation, which may play a more significant role than school attendance and coeducational status by contributing to HPV vaccination hesitancy [24,25,26].

While we had a large cohort of over 9000 students at 70 schools, our study may have lacked power to identify significant associations for all factors, particularly for HPV coverage, due to the fewer number of schools when analysing by gender.

The ecological study design limits causal inferences and does not account for individual-level factors that may influence vaccination uptake. Furthermore, Aboriginality and LBOTE data were only available at the school level rather than the year level, which may impact the accuracy of our analysis. Finally, the focus on southeastern Sydney may limit generalisability, and reliance on school records may underestimate coverage by excluding catch-up and GP-administered vaccinations.

### 4.5. Implications and Future Research

Our findings have important implications for policy and practice in school-based vaccination programs, highlighting the need for targeted interventions tailored to individual school characteristics. Key policy implications include developing targeted interventions for schools with lower attendance rates to boost vaccination coverage; implementing strategies for coeducational schools to improve HPV vaccination rates; continuing to monitor equity in vaccination access across different socio-economic backgrounds; and considering gender-specific approaches within mixed settings to create more supportive environments for vaccine administration.

Future research should focus on exploring the mechanisms behind the observed associations through mixed-methods studies, particularly differences between single-sex and coeducational schools; examining qualitative aspects of school environments influencing vaccination uptake; evaluating intervention strategies across different school types; and further investigating the utility of ICSEA in examining vaccination disparities. Other factors that require a different research approach to determine, such as school organisational capacity, effectiveness of communications between the school and parents, and access to alternate vaccination providers, may all influence variation in uptake between schools.

It would be useful to conduct a similar analysis at the state level to determine if these factors are applicable to other locations and have more power to elucidate the significance of school sector and LBOTE. Such an approach could identify regional patterns and inform a coordinated statewide strategy, enhancing the effectiveness and equity of school-based vaccination programs across diverse contexts. These directions aim to deepen our understanding of factors influencing school-based vaccination programs and inform the development of more effective, targeted strategies to improve vaccination coverage among adolescents.

## 5. Conclusions

This study moves beyond average vaccination rates to explore disparities in dTpa and HPV vaccination coverage among schools in southeastern Sydney. Our findings highlight the significant roles of school attendance and coeducational status in influencing vaccination coverage among year 7 students. School sector and LBOTE may also influence coverage and are worthy of further exploration. The differential impact of these factors on dTpa and HPV vaccination underscores the complexity of influences on adolescent immunisation. However, recognising the complexity of factors at play, further research, including qualitative approaches, is essential to unearth additional factors influencing vaccination coverage in high schools and to devise comprehensive strategies for improved and inclusive coverage. This study underscores the importance of nuanced, context-specific approaches to school-based vaccination programs.

## Figures and Tables

**Table 1 vaccines-12-00888-t001:** Characteristics of schools and vaccination coverage, SESLHD School Vaccination Program, 2019.

	*dTpa*		*HPV—Girls*		*HPV—Boys*	
	Number of Schools (%)	Median Vaccination Coverage (IQR)	Number of Schools (%)	Median Vaccination Coverage (IQR)	Number of Schools (%)	Median Vaccination Coverage (IQR)
*LBOTE*						
<=25%	29 (41.4%)	87.3 (82.5–91.5%)	23 (41.1%)	88.2 (84.5–91.8%)	21 (41.2%)	86.3 (79.3–88.6%)
>25%	41 (58.6%)	90.2 (81.6–94.7%)	33 (58.9%)	88.6 (84.2–91.5%)	30 (58.8%)	87.0 (77.6–92.2%)
*Aboriginal enrolments*						
<1%	44 (62.9%)	90.4 (85.3–94.0%)	33 (58.9%)	87.7 (84.2–91.5%)	31 (60.8%)	88.2 (82.6–92.2%)
≥1%	26 (37.1%)	86.0 (79.1–92.4%)	23 (41.1%)	89.3 (84.3–92.1%)	20 (39.2%)	80.2 (74.9–88.7%)
*Attendance*						
Semester 1 attendance <=92%	31 (44.3%)	84.7 (76.9–90.1%)	27 (48.2%)	86.3 (78.6–90.2%)	26 (51.0%)	82.5 (74.4–89.6%)
Semester 1 attendance >92%	39 (55.7%)	91.3 (85.9–94.0%)	29 (51.8%)	90.4 (85.3–92.6%)	25 (49.0%)	88.2 (82.9–92.2%)
*Coeducation status*						
Coeducation	37 (52.9%)	85.7 (78.3–91.0%)	37 (66.1%)	85.7 (80.0–90.4%)	37 (72.5%)	84.3 (77.4–89.2%)
Single sex	33 (47.1%)	91.3 (87.3–94.3%)	19 (33.9%)	90.9 (88.5–93.1%)	14 (27.5%)	90.1 (83.6–93.9%)
*School sector*						
Government	33 (47.1%)	86.7 (85.5–87.9%)	28 (50.0%)	86.7 (82.9–91.9%)	27 (52.9%)	85.9 (74.9–91.3%)
Independent	22 (31.4%)	86.2 (81.5–91.3%)	18 (32.1%)	86.5 (82.6–91.3%)	15 (29.4%)	83.3 (83.2–83.4%)
Catholic	15 (21.4%)	92.5 (90.6–94.5%)	10 (17.9%)	91.2 (89.8–92.7%)	9 (17.6%)	89.2 (88.2–93.3%)
*ICSEA*						
High (1100+)	27 (38.6%)	90.7 (85.7–94.0%)	20 (35.7%)	90.5 (84.7–92.3%)	17 (33.3%)	88.1 (82.2–93.3%)
Medium (1000–1099)	36 (51.4%)	86.6 (80.8–91.4%)	30 (53.6%)	87.5 (84.2–91.4%)	28 (54.9%)	83.6 (77.0–88.4%)
Low (<1000)	7 (10.0%)	88.6 (76.0–95.0%)	6 (10.7%)	87.7 (63.9–89.2%)	6 (11.8%)	90.5 (88.9–92.4%)
Total	70 (100%)	88.2 (81.9–93.0%)	56 (100%)	88.4 (84.2–91.6%)	52 (100%)	86.3 (78.3–91.3%)

**Table 2 vaccines-12-00888-t002:** Adjusted vaccination coverage prevalence ratios (PR) by school characteristic SESLHD School Vaccination Program, 2019.

	*dTpa*	*HPV—Girls*	*HPV—Boys*
	Adjusted Prevalence Ratio (95% CI)	Adjusted Prevalence Ratio (95% CI)	Adjusted Prevalence Ratio (95% CI)
*LBOTE*			
<=25%	1.00	1.00	1.00
>25%	0.99 (0.94–1.05)	0.91 (0.82–1.01)	1.02 (0.92–1.13)
*Aboriginal enrolments*			
Aboriginal enrolments <=1%	1.00	1.00	1.00
Aboriginal enrolments >1%	1.00 (0.94–1.05)	0.98 (0.89–1.07)	1.06 (0.95–1.19)
*Attendance*			
Semester 1 attendance <=92%	1.00	1.00	1.00
Semester 1 attendance >92%	1.14 (1.02–1.27)	1.14 (0.93–1.38)	1.09 (0.91–1.31)
*Coeducation status*			
Coeducation	1.00	1.00	1.00
Single sex	1.03 (0.98–1.09)	2.24 (2.04–2.46)	1.89 (1.72–2.08)
*School sector*			
Government	1.00	1.00	1.00
Catholic	0.97 (0.87–1.08)	0.93 (0.77–1.13)	0.97 (0.81–1.15)
Independent	0.91 (0.83–1.01)	0.86 (0.72–1.02)	0.94 (0.80–1.11)
*ICSEA*			
High (1100+)	1.00	1.00	1.00
Medium (1000–1099)	1.00 (0.91–1.10)	1.04 (0.88–1.22)	0.86 (0.73–1.01)
Low (<1000)	1.07 (0.94–1.21)	0.89 (0.71–1.12)	1.18 (0.96–1.46)

## Data Availability

The datasets presented in this article are not readily available because of privacy restrictions. Requests to access the datasets should be directed to the corresponding author.

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
