# Peer review of "Beyond Averages: Unpacking Disparities in School-Based Vaccination Coverage in Eastern Sydney: An Ecological Analysis"

_vaccines, 2024, doi:10.3390/vaccines12080888_

Round 1

Reviewer 1 Report

Comments and Suggestions for Authors

The manuscript contains several interesting pieces of information, but some points can be improved:

- Expand the discussion on the contrasting results between the dTpa vaccine and the HPV vaccine, especially in relation to the association with school attendance. Explain in more detail why school attendance had a significant impact on dTpa coverage but not on HPV coverage.

- Discuss the possibility of other unaccounted variables, such as parents' perceptions of vaccines, school communication about vaccination, and the availability of health resources, which may influence vaccination coverage.

- Consider further disaggregation of the data by subgroups, such as different age ranges or income groups, to gain a more detailed understanding of disparities.

- Include a sensitivity analysis to test the robustness of the results by varying the cut-off points for the classified variables (e.g., school attendance, ICSEA).

- Provide a more detailed interpretation of the confidence intervals of the odds ratios, highlighting the uncertainty associated with the estimates and how this affects the study's conclusion.

Comments on the Quality of English Language

Minor editing of English language required

Reviewer 2 Report

Comments and Suggestions for Authors

This manuscript is an interesting ecological study in schools that deals with vaccination coverage. I think that is solid from the methodological point of view, and I should be published—some questions reamin.

MAJOR ISSUES

1)      Please incorporate   Directed Acyclic Graphs (DAGs) into your analysis to elucidate further the complex relationships between school characteristics and vaccination coverage. .( Utilizing free software like Daggity  https://dagitty.net/)  DAGs can provide a clear visual representation of these relationships, assist in identifying potential confounders, and improve the robustness of causal inferences. Many examples of DAGS with Dagitty pub are published in MDPI’s Journal.

Minor issues:

 2)      Please consider that the paper will be read by people unfamiliar with the Australian healthcare and education sector.

3)      Abstract:  The first time the abbreviation ICSEA is introduced in the abstract, what it is should be explained.

4)      Introduction: Explain in a short paragraph what is ICSEA

5)      Explain in more detail whether the vaccine is free for all participants or if they have to pay for it.

6)      Material and Methods: Explain why you use quasipossion models. Usually, the reason is the existence of overdispersion. Is this the reason? How did you measure it?

7)      Explain how ICSEA was computed. Was it computed by the authors? Did it come from a database ¿ How did they do it? It should be explained.   (I think this information is detailed in their methods section, where they mention integrating vaccination records with a dataset provided by ACARA using the ACARA ID as the linking variable, but it is unclear.)

8)      Did you use an R library or module or command for the analysis? Please, please mention it.

9)      Did you use additional software with R, like R Studio or R Commander? Please mention it.

10)  What does it mean? School sector ¿ Is a geographical area or is a type of administrative classification for funding?

Reviewer 3 Report

Comments and Suggestions for Authors

The manuscript "Beyond Averages: unpacking disparities in school-based vac-cination coverage in eastern Sydney - an ecological analysis" is a valuable contribution. It summarizes an investigation of school-specific factors influencing the uptake of adolescent universally recommended vaccines in eastern Sydney. The study is well introduced, the methods are carefully described, and the results clearly presented. 

As authors mentioned, it is not possible to infer on causal associations based on an ecological analysis. However, taking into the limitations of the analysis, I think it's a good example of hypothesis-generating research, that needs to be followed by thorough qualitative studies and will be a good basis to improve the adolescent immunization programme in this region. I recommend publication of this manuscript since it adds evidence to the global discussion, and is a nice example of valid, reproducible operational research.

I just have few minor suggestions for improvement:

1. In the paragraph before Table 1, the authors mention that some schools had very low uptake (31% for dTpa, 25% for female HPV and 35% for male HPV. Maybe they could add a paragraph describing these outliers (maybe some disting characteristics of these schools could provide some context to their discussion on significant associations). To avoid problems with confidentiality, maybe they could describe the lower quintile?

2. There were some problems with formatting - the last two paragraphs of the background and in few other places, the font was decreased. 

Reviewer 4 Report

Comments and Suggestions for Authors

Well written manuscript. 

Need clarification on:

1. It is not clear to a non-Australian reader how the schools are classified using the ICSEA scores. 

2. In the geographic area chosen, how many schools fall under the 3 categories described - high, intermediate and low?

3. Is it right in assuming that the predominant classifiers are - aborigines, non-English speakers and below a percentage of income, possibly, the 3 characteristics of people of European, non-Asian origins.  

4. It would be important to know what percentage of total school children should be of aboriginal origin to be classified as high, intermediate and low.

5. Is there any precedent for your surmise to choose ICSEA scores as indicative of a pupil's health, in any study from Australia.

Round 2

Reviewer 1 Report

Comments and Suggestions for Authors

Citations and references are inconsistently formatted. For example, some references are numbered (e.g., [119], [120]), while others are mentioned by name and year (e.g., Marra et al., 2020). Furthermore, the use of commas and periods within quotations is not uniform.

There are several typographical errors, such as the absence of spaces between words ("viralreplication") or duplication of words ("the the"). These errors can distract the reader and should be corrected.

Some parts of the text are redundant or not well structured, which can make understanding difficult. For example, the repetition of phrases such as "In Vivo Efficacy" and "Mouse Model" without a clear logical progression can confuse the reader.

Tables and figures are not clearly identified or referenced in the text. For example, reference to "Figure 4" without a clear description or associated caption may make it difficult for the reader to understand.

Some parts of the text do not adequately cite the sources of the information presented. The inclusion of appropriate references is essential to validate the claims made and allow readers to consult the original sources.

Comments on the Quality of English Language

Needs adjustment, with several problems in relation to understanding the text

Author Response

Please see attachment - we are concerned that not all the comments relate to our paper.

Reviewer 2 Report

Comments and Suggestions for Authors

The authors have satisfactorily answered all the questions posed to them

Author Response

Thanks again for your very useful suggestions.